# Documentation Status and Youth's Critical Consciousness across Borders

**Sarah Gallo [1],\* and Melissa Adams Corral [2]**

1. Graduate School of Education, Department of Learning & Teaching, Rutgers University, Newark, NJ 08901, USA
2. Department of Teaching and Learning, University of Texas Río Grande Valley, Edinburg, TX 78539, USA
* Correspondence: sarah.gallo@rutgers.edu

**Abstract:** Centering the testimonios of two sets of transborder high school seniors living and learning in Mexico and the U.S., in this article, we draw upon decolonizing approaches to theorize critical consciousness formation for and with students from families with mixed documentation status who cross physical and metaphorical borders. Data come from two larger qualitative studies on immigration and education and demonstrate how young people recognize inequity, critique it, and engage in a range of actions to counteract it. We argue that border-crossing youth draw upon personal experiences to critique and take action to change oppressive realities. We extend critical consciousness scholarship by bringing unique attention to the role of undocumentedness in critical consciousness formation.

**Keywords:** critical consciousness; transborder; documentation status; testimonio

## 1. Introduction

> I think that fear never really went away. It's just something that I got used to. Like even now it becomes very scary when the police stops us, and they give us a ticket . . . Like even my brother—and he was born here—he is scared. I think we all just get that, I guess that heartbeat, you know? And like it's the fear like, 'oh my God, they're going to take him away,' you know? Like, what's next? I don't think that ever really goes away. But my parents and I have a plan, and we talked about it for many, many years. And as I grew up, I knew exactly what to do. And if that were ever to happen, I am not necessarily scared . . . My mom always told me, she was like 'sell all our stuff, buy the tickets, and then just go with your brother', you know? And it was just, 'Go to Mexico. We'll meet you there'.—Abi Interview, 13 January 2021.

This quote comes from Abi, a high school senior living in Pennsylvania, whose life unfolded across physical and metaphorical borders as a member of a mixed-status family, where only some members have access to U.S. documentation (Fix and Zimmerman 2001). Through her reflection on living with the constant threat of potential deportation, she shared experiences related to critical consciousness formation, or the ability to recognize, critique, and take action against systems of inequity. Abi had intimate experiences with inequity: she and her family members felt in constant danger when dehumanizing local immigration policies transformed routine practices, such as driving, into unlawful acts for those who had no pathway to U.S. papers (Gonzales 2016). Abi did not just notice these inequities—she also became increasingly critical of them, naming the unnecessary stress they caused her and her brother throughout their childhood. Abi also shared the plan of action if her parents were ever deported: she would sell their belongings and travel with her brother to Mexico where they could reunite as a family. As in many mixed-status families, increasing threats of deportation during the Obama, Trump, and Biden administrations

shaped the contours of Abi's childhood, while Abi engaged in action to combat these inequitable systems through collective plans for action.

Drawing from two qualitative studies with mobile students in the U.S. and Mexico, in this article, we explore transborder youth's experiences across borders to theorize critical consciousness with and for young people navigating undocumented status in their families. The first study, an ethnography conducted in the homes and schools of students who had moved from the United States to their parents' hometowns in Puebla, Mexico, highlighted experiences with inequities due to U.S. immigration policies, such as family separation, social and academic exclusion, and constrained movement across borders without U.S. papers. The second study was an in-depth interview study with high school seniors in Pennsylvania from mixed-status Mexican-origin families who had previously participated in a multi-year ethnographic study with Sarah during elementary school. Young people from this study navigated inequities such as managing invisibility from government authorities to evade familial deportation alongside hypervisibility to access higher education or employment (Gallo and Adams Corral 2023).

To understand how border-crossing youth draw upon their subaltern experiences to recognize and potentially combat inequity, here we investigate the following question: What experiences do transborder youth have with recognizing, critiquing, and taking action against systems of inequity related to undocumented status? We argue that transborder students, especially those directly experiencing inequitable realities, draw upon personal experiences to critique and take action on various scales to change oppressive realities in their families, schools, and communities. In what follows, we define our transborder critical consciousness framework, situate it within the relevant scholarship, and offer illustrations of critical consciousness formation in transborder youth. We extend critical consciousness research in three ways. First, we highlight the under-researched topic of how undocumentedness shapes young people's critical consciousness formation. Second, we prioritize decolonizing theoretical framings and methodological approaches, in which we listen deeply to young people's testimonios to offer alternatives to presupposed labels and variables that are often used in quantitative critical consciousness research. Finally, we extend critical consciousness scholarship by researching beyond the single nation-state, adopting qualitative methodologies reflective of the true movements across borders that are part of many young people's lives.

## 2. Framework, Literature, and Methods

### 2.1. A Transborder Critical Consciousness Framework

We focus on the experiences of transborder youth whose lives and schooling intersect with the realities of undocumentedness—impeded access to official papers such as birth certificates, visas, or passports for belonging—on both sides of the Mexico-U.S. border. Rather than viewing mobile students and families as immigrants, framings that presuppose directionality, timescales of im/permanence, or goals of assimilation for belonging in a single nation, our research adopts a transborder lens (Dyrness and El-Haj 2019; Mignolo 2000). Transborderness is a decolonizing approach that normalizes multidirectional movement across physical and metaphorical borders and the resulting in-betweenness people experience as part of their everyday lives and actively counters the assumption of mononational lives as a norm (Anzaldúa 1987; Dyrness and Sepúlveda 2020; Mignolo 2000). Transborder people draw on marginalized experiences to build and utilize subaltern knowledges (Cervantes-Soon and Carrillo 2016; Mignolo 2000) for navigating institutions, borders, and hierarchies that typically exclude them and their ways of knowing. A transborder approach is well-suited to understanding the specific inequities of young people in families navigating undocumented status.

Critical consciousness has conceptual roots in Paulo Freire's (1970) theory for how people collectively raise their consciousness (conscientização) of how power operates in society (reading the world), critiquing hegemonic realities and moving towards action (praxis) to counteract oppressive norms, hierarchies, and institutions of inequity. Situated

in the struggle for liberatory education with adults, Freire's (1970) original theorizing of critical consciousness education emphasized centering the prevailing concerns in students' lives, engaging in reflective action-oriented dialogue, and student-teacher relationships where learning and expertise are multidirectional and shared.

Research regarding the roles and possibilities of critical consciousness in education has proliferated across various fields, with different disciplines operationalizing Freire's original theorizing to align with their prevailing research approaches and goals (see Heberle et al. 2020 for an overview). Much of this research has centered adolescents in fields such as human development, with Freire's theory of critical consciousness reconfigured to measure individuals' capacity for critical reflection (naming and analyzing inequality), political agency (a belief they can enact change), and critical action (engagement in activities that counteract oppression) (Watts et al. 2011). This robust body of research has unfolded almost exclusively with Black, Latinx, and White adolescents within the U.S. Using quantitative analyses from large-scale survey data; it seeks to understand how critical consciousness can be optimally fostered across discreet developmental stages (Heberle et al. 2020). Although this scholarship offers important insights into factors shaping US-based teenagers' critical consciousness development, it largely relies on large-scale survey data that analyzes pre-determined variables, which may not capture embodiments of critical consciousness that do not map onto those established categories. This article addresses these limitations by offering explicit attention to how undocumentedness shapes young people's critical consciousness formation as expressed in students' testimonios.

Within the fields of bilingual and immigrant education, scholars have deployed Freire's (1970) theorizing of critical consciousness with K-12 students within decolonizing approaches that study the naturalization and reproduction of power hierarchies within schools and the resulting dismissal of the knowledges and resources of racialized students and families (Palmer et al. 2019). Decolonizing approaches question the colonial legacies of the hegemonic knowledges celebrated, normalized, and perpetuated in most schools and instead seek to center the subaltern knowledges developed by those who have been historically marginalized (Cervantes-Soon and Carrillo 2016; Mignolo 2000). Most of the re-search has explored younger children's bilingual educational settings in the U.S. and urges the adoption of critical consciousness as a pillar for program design, pedagogy, and power hierarchies (Cervantes-Soon et al. 2017; Heimen and Yanes 2018; Palmer et al. 2019). This scholarship, however, has centered on adults' roles in implementing critical consciousness schooling and offers less insight into students' experiences with critical consciousness in their own right. We build upon this by centering transborder students' perspectives and experiences related to critical consciousness formation.

In summary, situated in the fields of bilingual and immigrant education across geopolitical borders, in our scholarship with transborder students whose lives unfold across Mexico and the U.S., we orient to decolonizing approaches of Freire's (1970) theorizing of critical consciousness, which we operationalize into three parts. We explore (#1) how young people read the world or their experiences with inequity, (#2) how they critique inequitable systems, and (#3) how they plan or act to counter these inequities. Our ways of understanding transborder youth's actions (#3) are intentionally inclusive of interpersonal action (such as an individual's response to inequity), communal action (such as collective efforts to combat inequity in school, family, or community), and political action (such as political organizing) to broaden our gaze of how young people may counteract inequities (see Aldana et al. 2019, for similar conceptualizations of action). An inclusive definition of action also recognizes that, as members of undocumented families, political action related to voting or official political processes—from which their adult family members are often excluded—may not be the primary types of political action taken up by transborder youth (Diemer and Rapa 2016; McWhirter et al. 2019). This allows us to learn from transborder students' engagements with action on their terms rather than through pre-determined categories that may not be reflective of their realities.

## 2.2. Literature Review

### 2.2.1. Schooling across Borders

Mexico and the U.S. have one of the most expansive histories of immigration and schooling, including multidirectional movement across their shared border (Gándara 2020). Although this migration is often assumed to move from South to North, since 2005, this has shifted, with more Mexicans returning to Mexico than arriving in the U.S. (Gonzalez-Barrera 2015). It is estimated that there are at least 600,000 transborder students with US schooling experiences enrolled in Mexican schools (Jacobo and Jensen 2018). These students are a relatively invisible population, and Mexican schooling offers limited support for transborder students who bring different knowledges and linguistic skills (Bybee et al. 2020; Dreby et al. 2020; Kleyn 2017; Gándara 2020; Zúñiga et al. 2008).

Within US contexts, research with transborder students—especially around inequities related to undocumentedness—has brought important attention to how transborder students navigate schooling and life transitions as undocumented, DACAmented, or U.S. citizen children (Gonzales 2016). This research within high schools has highlighted how educators proactively integrate, broach, or evade topics related to immigration (de los Ríos and Molina 2020; Jefferies and Dabach 2014). At the elementary school level, scholars have illustrated how family members' undocumented status shapes young children's learning (Gallo 2014, 2017; Mangual Figueroa 2017) and the importance of creating a range of learning opportunities so that students can draw on their immigration experiences for learning if they choose (Gallo 2014; Mangual Figueroa 2017).

### 2.2.2. Critical Consciousness Research

Research on critical consciousness has shown that young people who have experienced structures of inequity, such as racial discrimination in the U.S., often demonstrate more nuanced critical consciousness development compared with their mainstream peers who often benefit from systems of privilege (Diemer and Li 2011; Kelly 2016; Seaton 2010; Seider et al. 2019). In previous collaborative work (Dreby et al. 2020), we explored the perspectives of transborder students attending elementary school in Mexico and highlighted the ways that their experiences with migration status shaped their understandings of inequity and how they managed this among peers. Research has demonstrated how critical consciousness development among marginalized students can serve as a protective factor against further marginalization by supporting them to engage in critical reflection and collective action to "overcome pervasive myths" (Cervantes-Soon et al. 2017, p. 419), blaming marginalized populations to instead center alternative histories revealing how hegemonic structures created oppressive conditions (Cammarota 2007; Ginwright 2010).

Research has also demonstrated that parents sharing critiques of inequities and potential responses to them with their children can foster greater critical consciousness development, especially in critical reflection (Bañales et al. 2020; Diemer and Li 2011). As Heberle et al. (2020) highlight, less is known about the possibilities of action-focused socialization, such as children witnessing or co-participating in political organizing. Transborder scholars have demonstrated that children and families share and develop opportunities for children to recognize and critique inequities central to their lives and highlight forms of political action engaged in by families such as migration, testimonios to counter oppressive structures, community organizing and protest, and self-determination to navigate the world as a member of an undocumented or mixed-status family (Gallo 2016; Mangual Figueroa 2013; Rusoja 2022; Salazar Pérez and Saavedra 2017).

Scholars, particularly in the fields of anthropology and education, have argued that youth whose lives unfold across borders are well-positioned to develop critical consciousness because of their intimate experiences with inequity and movement across contrasting national systems where their knowledges, languages, and practices are (mis)understood (Cervantes-Soon 2017; Dyrness and El-Haj 2019; Dyrness and Sepúlveda 2020). Dyrness and El-Haj (2019, p. 169), for example, argue that transnational youths' unique forms of "critical consciousness emerges from occupying a space between nations, with frames of references

that encompasses multiple groups and experiences across borders . . . including, all too often, negative judgements from both their home and host countries". Cervantes-Soon's (2017) ethnographic research with high schoolers in the borderlands of Mexico reveals their embodied awareness of the entangled oppressive systems that shaped their lives. Although this scholarship offers important insights on how subaltern border-crossing experiences can relate broadly to critical consciousness formation, they offer less specificity in defining critical consciousness and do not demonstrate how youths' specific practices map onto different components of critical consciousness formation.

In summary, development-oriented research offers specific operationalized definitions of critical consciousness and large-scale measurements of how adolescents engage in these components of critical consciousness development, yet this research rarely focuses explicitly on students navigating inequities related to undocumentedness. In addition, some of the measures traditionally used (e.g., around political engagement) may not fully account for students from mixed-status or undocumented families (Diemer and Rapa 2016; McWhirter et al. 2019). This body of research unfolds almost exclusively within the U.S. and tells us little about how young people's experiences with the crossing of geopolitical and metaphorical borders shape their critical consciousness. Decolonizing approaches to critical consciousness and education offer important insights into how students navigate inequities across borders and discuss critical consciousness in abstraction but rarely offer specific definitions or analyses documenting how these experiences map onto specific components of critical consciousness. Here we build on this research to closely analyze how youth from mixed-status families living in Mexico and the U.S. relate their experiences to the varying components of critical consciousness: noticing inequity, critiquing inequity, and taking action to counter systems of inequity.

### 2.3. Methods

### 2.3.1. Study Contexts

We focus on the perspectives and experiences of transborder youth from mixed-status families on both sides of the Mexico-U.S. border as they prepared to graduate from high school, a pivotal moment when undocumentedness—theirs' or their family members'—profoundly shaped their experiences as they transitioned toward adulthood (Gonzales 2016). Here we draw from two larger qualitative studies with transborder students from mixed-status families, one conducted in Mexico and one in the U.S., to center the perspectives and experiences of high school seniors whose lives and learning were shaped by immigration experiences.

The first study was conducted in Puebla, Mexico, in 2016–2017 to understand the interlocking realities of immigration and education policies for families who relocated to their parents' hometowns in Mexico due to U.S. immigration policies (Gallo and Ortiz 2020). From that larger yearlong ethnographic study across students' classrooms, communities, and homes, we analyzed in-depth interviews with three high school seniors from mixed-status families now living in Mexico. These students attended a small, rural high school of approximately 100 students with a single senior class located in an 8000-person town where almost everyone had family living in the U.S. without official documentation. The three students interviewed were from transborder families that had lived in or around New York City; one was a U.S. citizen, and the other two were not. The study was conducted during the year Trump was elected president, and students regularly discussed their understanding of living across borders within this heightened anti-Mexican and anti-immigrant context.

The second study was conducted in Pennsylvania in the United States in 2021 as an extension to a 3-year ethnography across home and elementary school contexts with six children from mixed-status families (Gallo 2017). The follow-up study in 2021 consisted of in-depth ethnographic interviews with each of the original children, now high school seniors, and explored their experiences as members of mixed-status families. Students from this study had attended elementary school during the Obama administration's era of heightened deportations, high school under the Trump presidency, and graduated

amid the COVID pandemic. Four of the students were U.S. Citizens, and the other two were applying for Deferred Action for Childhood Arrivals (DACA) during a period of continual precarity (Gonzales et al. 2019). Two students' parents had been deported from the U.S., one successfully fought her father's deportation, and the remaining three also had undocumented parents. Both groups of young people in these studies lived far from the physical U.S-Mexico border, and together their testimonios capture a range of transborder students' experiences navigating and contesting inequities related to undocumentedness.

### 2.3.2. Data Collection and Analysis

To understand how youth's experiences with undocumentedness shaped their critical consciousness, we draw upon the decolonizing approach of testimonios, an interactive narrative research practice that sanctifies space for the subaltern experiences of people who are often silenced or delegitimatized (Mangual Figueroa 2013; Solórzano and Yosso 2002). Testimonio, originating from oral traditions of the Latinx diaspora (Delgado Bernal et al. 2012), means to give testimony to others in order to push against the authority of traditional knowledge hierarchies to reveal injustice and engage in acts of collective resistance (Cervantes-Soon and Carrillo 2016). Interviews with participants offered opportunities for critical listening and a mutual sharing of selves that pushed against the rigid boundaries of traditional qualitative interviews, situated in long-term humanizing relationships of care between Sarah and participants (Paris 2011). The interviews in Mexico (N = 3) occurred face-to-face in students' schools or homes, and due to social distancing during the global pandemic, interviews in Pennsylvania (N = 6) occurred via videoconference. All interviews unfolded in students' preferred languages (English, Spanish, and translanguaging across both) and ranged from one to three hours in length, with most lasting about two hours.

Our analyses iteratively drew patterns from the data to center on meaningful themes in transborder students' lives. Monthly in-process memos during each study provided further direction during data collection through an examination of topics such as "deportation", "college access without papers", and "Trump in school". Once data collection was complete in each study, the interviews were transcribed by a local bilingual transcriber in each context. Each study was coded using online qualitative data analysis software (Atlas.TI), using separate code books. The codes from each separate study came from both etic (theories such as "transborder") and emic (participants' wording such as "papeles") approaches. The sub-corpus of data from the larger studies centered on transborder high school students' perspectives and experiences comes from codes such as "documentation" and "deportation" from the Pennsylvania study and "papeles", "children's diasporic relocations", "critique", and "subaltern knowledges" from the Mexico study.

### 2.3.3. Subjectivities

We are committed to humanizing research approaches that center on reciprocal relationships while questioning dynamics of power and privilege, challenging inequities, and contesting ideals of objectivity (Paris 2011). Sarah is a queer bilingual white woman from the U.S. who forms part of a mixed-status transborder family in which we live and learn, separated and apart due to differential access to U.S. papers across the Mexico-U.S. border. Participants were familiar with Sarah from her weekly participation in their educational lives at school and home for one (in Puebla) to three (in Pennsylvania) years. Melissa is a Latina whose transnational childhood included schoolyears in Miami and summers in Tegucigalpa with extended family. Her own experiences meant that as a young child, she was aware of the disparities of power, access, and comfort between her two nations, an awareness that helped spark an ongoing and active commitment to immigrant families as a bilingual teacher and researcher. Similar to our approaches as educators working with transborder students, in our research, we invited young people into conversations about undocumentedness if and when they were open to sharing and did not share their personal experiences with others in their schools or communities or in personally identifiable ways in our scholarship (see also Oliveira and Gallo 2021). Our experiences shape our scholarly

goals of recognizing the practices and knowledge exhibited by transborder youth and are founded in commitments to freedom of movement across borders as a basic human right.

## 3. Findings

In this section, we analyze transborder high schoolers' experiences with inequity related to documentation status on both sides of the border. We analyze examples of how students in Pennsylvania and Puebla experience and critique these inequities and then demonstrate the range of actions young people engage with, including supporting others in similar positions, raising awareness on an individual and public scale, and finding ways to continue to navigate the complexities of the immigration system in search of access to a more complete set of rights.

### 3.1. Youth's Experiences with Inequity

3.1.1. Fearing Deportation

As we saw in Abi's opening quote, high schoolers from mixed-status families in Pennsylvania shared intimate experiences with the inequity of government-sanctioned family separation through deportation. Throughout their childhoods, they were aware that their undocumented parents could be deported at any moment. U.S.-born Maritza reflected, "it's always in the back of my mind—what if for some odd reason some immigration police officer just comes to the door and asks, 'Where are [parents' full names]'? And if I'm their daughter, I am still 17, I could be in a foster home, that could happen. That's always in the back of my mind. I'm always questioning where my mom's at because, 'Where is she? And where is my dad? Is he okay?'." Maritza's family had not shared a deportation plan with her, and Maritza worried the forced removal of her parents would lead to her removal from her family and placement in foster care, demonstrating her awareness of the reverberating consequences following parental deportation. Beyond this understanding, Maritza describes the persistence of this fear—a constancy that reflects the context where children had been systematically separated from their caregivers at the border during the Trump Administration and, at times, stolen from their families and placed in adoption. Brittany, who had arrived in Pennsylvania without papers before her first birthday and, at nine years old, had witnessed her father's violent arrest when immigration officers mistook him for someone else, shared similar concerns, noting, "I could lose my parents in an instant . . . living in fear, that's one of the things that have impacted my life". Most of the Pennsylvania high schoolers noted how, as young elementary schoolers, they were worried that the police could take their parents away without fully understanding why. Over time they learned it was because their parents were born in Mexico and were unable to access U.S. documentation.

3.1.2. Differential Access to Documentation

Students, especially those with siblings of differing documentation statuses, also shared their awareness of inequities in movement and access related to documentation status. Samuel, a senior on the brink of graduation from his rural high school in Mexico, reflected on the different pathways he and his U.S.-born brother could take post-graduation, reflecting, "I hope that he betters himself. Yeah, maybe he'll be better than me and all that. And I would like him to go to the U.S., because, well, I'd say that work here is really hard". Although Samuel's brother had moved to Mexico before starting elementary school, Samuel felt their parents had supported his brother's academics more because he could easily return to the U.S. for college or employment. Samuel's classmate in Mexico, Aaron, shared similar sentiments of inequity—that his parents let his U.S.-born brother get away with an apathetic work ethic because he could easily be employed in the U.S. and earn far more money than family members in Mexico or in the U.S. without documents. Back in Pennsylvania, Abi also reflected on how her U.S.-born brother could spend summers with their beloved grandparents in Puebla, whereas she could not because she lacked access to a U.S. passport that facilitated returning to the U.S. While transborder students were

aware of these inequities, they did not express jealousy or negative feelings toward their U.S.-citizen siblings—they were happy for their opportunities and longed for similar rights themselves.

### 3.1.3. Family Separation

In addition to the constant fear of potential family separation due to deportation, many of the young people interviewed shared the challenging inequities they had lived with when borders separated them from their parents. Emily was born in Pennsylvania, and during the summer after second grade, her undocumented mother returned to Mexico to bring her older daughter to live with them in the U.S. During their recrossing, Emily's sister made it to Pennsylvania, but U.S. immigration authorities apprehended her mother. Emily reflected on the challenging family dynamics over the following five years as her family fought inequitable systems to be reunited:

> It was definitely really, really hard, because I didn't understand for a while. I couldn't understand why my mom couldn't come back... It was really hard when I went back into third grade, because my mom was always the one that would get me ready for school. Something as simple as school supplies was something that I didn't even know how to get. Then when my sister finally came, it was also really hard because we used to not get along. It took a while for us to really get along and actually start acting like sisters. It was also really difficult when my mom did come back in seventh grade, which was something that I did not expect. I thought that when she came back, everything would go back to normal and that everything would be easier. But instead, it wasn't how I pictured it. It was hard for us to get used to having to listen to her rules and not do what we used to do when it was just my dad and my sister and I, so that was something that I didn't expect. Yeah, it was hard, but I think that it helped me to grow as a person.

One of Emily's classmates in Pennsylvania, Princess, faced similar inequities in second grade when her father was deported to Mexico. When he was caught attempting to return to the U.S., he was incarcerated for several years and then relocated permanently to Mexico, extending her family separation across geographies and timescales. Samuel, in Puebla, also described the challenges of desperately missing his mother when he was in Mexico under his grandmother's care and his surprise at how challenging it was to get along after his mom returned to Mexico. As immigration scholars have shown, children's separation from their parents due to immigration policies causes challenging chasms that take time to remedy if and when they are reunited (Dreby 2015; Oliveira 2018). Similar to Emily, Samuel spoke about how these difficulties forced him to grow up, explaining how he would tell his mother, "I am grateful that you sent me somewhere else, because thanks to you, I learned". These examples allow us to see how youth understand the awareness that comes from having experiences as young children that are deeply unfair, but that have also shaped whom they grew up to be. Awareness of these inequities offers opportunities for youth to formulate critiques about the systems shaping their lived realities.

### 3.2. Youth's Critiques of Inequity

### 3.2.1. Money or Family

Transborder students were not only aware of inequities related to documentation status but also had critiques to offer. Samuel, from a small town with a limited agrarian economy in Mexico, poignantly critiqued how systemic inequities within and across borders shaped most of their lives, "You basically never can have both things—you're with your family, or you have money, but never both together". He was aware and critical of how, in towns such as theirs, you could have a unified family but lack employment opportunities to meet basic necessities such as the associated costs to attend high school, or you could have material items through remittances from family members working in the U.S., but you could never actually spend time with them because you were separated by border policies. Samuel's classmate Alondra, who was born in New York and relocated to her

father's hometown during high school, often heard from educators that her life would be better if she returned to the U.S., where they imagined her having more educational and employment opportunities. However, she critiqued this assumption, explaining that she wanted to be where her family was even if they had few material belongings, which meant staying in Mexico. In this critique, Alondra pushed against assumptions of superiority of the global north (Dyrness and El-Haj 2019) and longed for a life with both a unified family and economic stability. For both Samuel and Alondra, inequitable immigration policies forced families to choose between economic instability together and relative economic security apart.

3.2.2. Blaming Systems, Not People

Transborder high schoolers reflected on the inequities related to undocumentedness and offered critiques of these realities, moving towards shaping their identification of systemic inequities. Emily, in Pennsylvania, was separated from her mother for five years due to undocumentedness and noted this shift in her perspective: "I feel like I kept thinking about our situation and I really didn't think about my mom's situation in the way that she was alone, that she didn't have us there [in Mexico]. I can't imagine how hard that must've been and literally, not being able to do anything about it. It wasn't her fault. It wasn't anyone's fault, which I think that was a really big thing because when things like this happen, you just want to find someone to blame for it. But who can you blame? It's not the family's fault, you know?" Emily knew that she and her mother's experiences were out of any individual hands—there was no "someone" to blame. For young people forming critical consciousness, this represents a crucial step towards making sense of the presence and impact of systemic inequities.

Transborder students living without papers in the U.S. were also critical of how their exclusion from formal national belonging shaped their lives. Brittany named how her exclusion from US papeles was particularly salient because it precluded her from health insurance that would support necessary gender-affirming surgeries. She explained, "I don't have a Social Security number. I wasn't born here. Because for surgery, I need a Social Security number so I can get insurance from the state, but I don't have one. So, I have to wait until I apply to DACA, before I can put my papers in and get the insurance". Brittany was clear that a complex web of systems lies between her and her basic human rights, understanding, similar to Emily, that this was an issue that went beyond blaming one person.

As Abi navigated the college application process without papers, she critiqued how as an undocumented "immigrant you really don't get too many privileges. And for example, college is one of them. Because they require a social security number. And then programs helping me with scholarships, I was asking for money, that was a big stab. And it's like, well, not being from here, it has its downfalls. Or getting a job, because part of growing up, it's like really hard. And that's I guess when it really hit me. Now I see the struggles that my parents went through, and I'm living them on my own". As young people in Roberto Gonzales' (2016) study demonstrate, Abi moved from seeing her parent's individual struggles as a child to now embodying and understanding how they were systematically influenced by exclusions related to documentation status.

Overall, students on both sides of the border were critical of the Trump administration and how rhetoric and policies emboldened anti-immigrant sentiment and exacerbated inequitable systems. Brittany, for example, noted how his administration was particularly harmful to "Mexicans because he deported a lot of people. He put kids in cages. He also took away LGBT rights...women's rights, and Obamacare . . . and everything in the capital last week [the January 6th attack]. Yeah, he's not the best president". Although Brittany named the president to represent shifting realities under his administration, she was aware of growing inequities that shaped her and her peers' lives in a range of ways. In Mexico, Samuel extended these critiques to explicitly name Mexico's systematic role in deportation regimes as well:

Central and South Americans have always said that the most difficult part is passing through Mexico, not the U.S. They've suffered more in Mexico, and more people (migrating) have died in Mexico than the U.S. And this is what you could say I don't like about my country [Mexico], the injustice. Here there is a lot of injustice, too much injustice. I mean we criticize the U.S., but we're really the ones who are more racist than the U.S. We kill people because they are not from here.

In 2017, long before major news outlets were highlighting the role the Mexican government was playing in deterring transborder populations from making it into the U.S., Samuel was critical of the role the Mexican government also played in upholding dehumanizing systems (Minian 2020). As Dyrness and El-Haj (2019, p. 169) demonstrated, experiences such as these provide opportunities for youth to "develop a critical awareness of the contradictions of Western democracy: the failed promises of inclusion and equality in their new state; and the ways that economic, political, and military policies of their host countries are often imbricated in the perpetuation of unjust, inequitable, and dangerous conditions in the countries from which they or their families emigrated". For Brittany, the U.S. repeatedly failed to provide inclusion and equality for several marginalized groups, while Samuel pointed to how this critique of the U.S. should not ignore the ways Mexico also perpetuates injustices. These injustices may include racism against migrants, but also the ways both countries work in tandem to create impossible forced choices—money or family (or safety or health)—that shaped the experiences of the young people in these studies.

### 3.3. Youth Take Action

In this final section, we highlight a range of ways that transborder students take interpersonal, communal, and political action to work against inequitable systems related to documentation status, forms of action that differ from the government-centric definitions for action that dominate much of the critical consciousness research with young people in the United States.

### 3.3.1. Interpersonal Action: Sharing Their Experiences to Reveal Inequities

One of the main ways transborder youth talked about taking action was through testimonio or sharing their experiences in everyday interactions with individuals who may not understand the systemic inequities related to undocumentedness. Brittany in Pennsylvania explained, "I don't really mind sharing, because it's like people could hear my story, and people could hear what I lived through, and the privileges they have that I don't have". Abi, in Pennsylvania, highlighted, "I don't have a problem actually talking about my life and my vulnerabilities, because I think that it would help a lot of students, in a way, understand and help them know more what we go through and so that they can open up". Unlike the 'don't ask, don't tell' de facto policies about undocumentedness that were part of most U.S. schools (Jefferies and Dabach 2014), transborder youth thought it was important to teach others about these experiences as part of their individual interactions, with Brittany seeing the value of more privileged individuals recognizing how systems that may benefit them are marginalizing others and Abi speaking to the potential for allowing others to share their own experiences at the margins. In both cases, these youth advocate for opening civic dialog to increase awareness, empathy, and unity.

Many students on both sides of the border also took action by helping new transborder students in their schools. Aaron, who had returned to high school in Mexico after seven years in U.S. schools, explained how he tried to help new kids, showing them around and translating for them if they did not know Spanish because "Here, there's people who look at you. The way you dress, the way you talk. Right away they start looking, well they start talking bad about you". Gregorio and Emily, both born in Pennsylvania with undocumented parents, tried to welcome new students and asked teachers to make school-based information more accessible for students who preferred Spanish, such as having bilingual homework assignments. Their experiences navigating these same inequities led to critiques that spurred their current actions at the community level.

### 3.3.2. Interpersonal Action: Disrupting Discrimination

Transborder youth also took action to interrupt discriminatory behavior. For example, Abi shared how a substitute teacher kept asking a student from a Chinese immigrant family racializing questions about his phenotype, study habits, and eating rituals in a public forum. Abi recounted, "and I remember that my friends and I just kind of got involved, and I was like, 'Well, I think that's a very stereotypical question for you to ask and I don't really feel comfortable with it'. I knew that he was uncomfortable, I felt uncomfortable. And she was like, 'Oh, no, I'm just asking because I want to know'". Rather than remaining silent about this teacher's discriminatory behavior, Abi interrupted it in the moment and later approached a teacher she trusted to report it as well.

Maritza, who worked in a local fast-food restaurant, reflected on her regular encounters with 'Karens', a term for entitled White middle-class women who racialize others in everyday interactions. She provided an example from that morning of a Karen who angrily complained about Maritza's poor service when she did not receive her catering order promptly, only to discover that the woman had made the order for the wrong date. Rather than remaining silent, Maritza drew upon a range of strategies to disrupt this woman's behavior, such as steadily disproving the woman's claims of poor service by telling her: "Ma'am, I did not give you one single bad attitude. You're the one who's causing the problem. If you wanted to place a catering order, you would have gotten it 20 min from now if you would have been more specific". When the woman requested Corporate's number to complain, in collaborative action with her manager, also from an immigrant family, Maritza instead gave her her own number, where Karen's unfounded complaint would go unheard. Transborder students such as Abi and Maritza were aware of systems of inequity in their everyday lives and sought to reach out to allies, such as trusted teachers or managers, to interrupt them rather than remain silent.

### 3.3.3. Communal Action: Planning against Deportation

Transborder students' actions also included strategic planning for the potential deportations of their parents. Maritza reflected on a lesson in her Pennsylvania high school class where her teacher asked her what she knew about immigration, leading her to realize she actually knew very little despite being from a mixed-status family. She reflected on how that moment sparked action:

> That day changed my life because ever since then I was like, "Oh, I need to focus, investigate more of what I need to do. What happens if a cop comes and he asks about my parents? What if we get pulled over one day and . . . it's just me and my mom?" What in the situation could I do? Should I know a good lawyer, even though we don't have the money for it? But we know who to call, one who's not scamming us. So we know who can we rely on. I learned about every possible way of avoiding anything to be separated because it's just me and them. And they're my only family. So, I can say that's the vivid moment I realized I needed to focus more of my parents' immigration status.

For Maritza, her action was about learning more and developing a plan if her family was put in danger. Abi, from the opening vignette, also planned for potential deportation, not just within her family but also through allies. She explained how her parents formally wrote a temporary custody letter for her godparents, U.S. residents, to extend their parental rights in the event of deportation. Abi explained how "they were part of the plan as well because they are residents and they are from here. So, it was just basically once we got the tickets, they would just allow us to go to Mexico because we couldn't go without the permission of an adult . . . I know exactly what to do". A key facet of transborder youth's critical consciousness involves the act of educating themselves on how to navigate the cruelties of U.S. immigration enforcement practices. Rather than seeing this as an adult concern, these youth recognize how planning, awareness, and strategizing skills may be key to keeping their family together and act accordingly.

### 3.3.4. Communal Action: Raising Collective Awareness

Transborder high schoolers also acted as part of formal groups to address systemic inequities related to undocumentedness. Both Emily and Abi in Pennsylvania were part of their high school's Dreamers group, designed for those affected by and interested in combatting inequities related to immigration rights. Emily explained that it gave her an outlet to talk about difficult topics and raise awareness. Abi highlighted being invited to talk to local educators about discriminatory practices towards immigrant students in their schools and encouraging teachers to always speak up and interrupt inequities in their classrooms. She shared how she advised them: "I think we all have questions of what to say and what is not to say . . . . It's fine (to talk about it), but it's not fine to be quiet . . . There's no reason to be afraid that you're going to say the wrong thing. We're all going to say the wrong thing, because we're all still learning". Transborder students such as Abi regularly engaged in testimonio, in which her personal narrative "becomes the means for agency through which the testifier not only has the opportunity to expose injustice and pain, but also to resist and counter dominant narratives, connect with others in profound ways through the confession of experiences, and to provide advice based on the wisdom gained" (Cervantes-Soon and Carrillo 2016, p. 292). Brave in the face of traditional power dynamics with her predominantly white middle-class teachers, Abi generously gifted others with insights into her subaltern experiences and the need to contest them. By doing so, Abi took her practice of testimonio from individual conversations into the public sphere and urged further actions from those in more powerful positions.

### 3.3.5. Political Action: Extending Rights to Documentation

In critical consciousness research, the focus on political action often names practices such as engaging in official elections, political organizing, or—more recently—engaging in protest (Heberle et al. 2020). However, as scholars have noted, these traditional practices may not align with the types of activities engaged in by young people from undocumented families (Diemer and Rapa 2016; McWhirter et al. 2019). To this end, we noticed that political action for many transborder students entailed working within current immigration systems to extend citizenship rights to others in their community. For example, Maritza shared how "I was trying to get a visa for my mom so she can see my grandma. And like a weight would be lifted from me knowing that they're relying on me to give them papers so they can go back after 23 years. And that's what's been my priority. That's what I have my money saved up for". Abi, who had a U.S.-born brother, similarly reflected, "His dream is like, 'Oh, I'm going to be 21 and I'm going to ask for my parents and I'm going to ask for you'".

In Mexico, Aaron, who did not have U.S. documentation, was also deeply knowledgeable of the intricate legal pathways to return to the U.S. Upon graduation, he shared how he was in the process of legally changing his last name to match his U.S.-born brother's, with the hopes that his brother could later sponsor him to live and work in the U.S. Young people were savvy and planned for changing legal requirements for visas and documentation, and also recognized these ever-changing processes took time. While it is common for many to assume that those who wish to access papers to live and work in the U.S. should simply 'follow the process' (Gonzales 2016), for transborder youth, there is an awareness that this process is more of an ever-shifting maze than a straight line. As such, those with U.S. documentation take action by seeking ways to support family members through strategizing around how to make the most of the set of papeles their family has been able to access.

Transborder students in Pennsylvania who were applying to college during the Trump and then Biden administrations saw DACA become a battleground, with presidential administrations and judges halting, re-starting, and stalling new and continuing applications (Gonzales et al. 2019). For US-based youth who arrived as young children, navigating the very unsteady waters of DACA was a form of action to extend documentation rights. Brittany, who sought DACA to continue her education and to access health insurance for

gender-affirming surgeries, explained, "when Trump was elected for president, I couldn't apply because I wasn't of age yet. So, I just had to wait until they put it back in". Abi, who intended to pursue a medical career in the U.S. or in Mexico if her DACA status was not approved, explained,

> And with the president (Trump), he canceled it. And now that it's back, the minute that they opened it again, we got my application, my information and it was just getting to know what were the requirements to get into DACA. And we had someone help us. She isn't a lawyer, but she helped start the DACA process because she did it as well . . . . And currently right now my application was accepted because they got the money. So now we just got to wait for it to be approved . . . As long as DACA still exists, it should be good—approved.

However, as she awaited the final steps for her application approval, DACA was re-frozen, and her plans to matriculate in a 4-year college to pursue a medical career were also stalled. Young people such as Abi and Brittany engaged in communal action with their peers, family members, and advocates versed in the ever-shifting realities of DACA requirements to extend their rights. What is striking about this particular set of actions is the way that undocumented youth continue to navigate a system that has repeatedly revealed its inequities and discriminatory nature to them. Despite that, they continue to seek and follow processes in the hopes of being recognized for whom they are—transborder youth seeking to live lives of their choice with their families and refusing to render themselves defeated.

## 4. Discussion

This article defines critical consciousness as awareness of inequities, critiques of these inequities, and moving toward action to combat them. Through the testimonios of high school seniors from mixed-status families on both sides of the border, we offer a unique window into how transborder youth draw upon their experiences with undocumentedness to engage in these various aspects of critical consciousness. We offer new insights into critical consciousness research by demonstrating the varied forms of action engaged in by transborder youth whose families are often excluded from traditional political systems, which differ from the forms of political engagement centered in most development-oriented critical consciousness scholarship. Action for transborder youth is interpersonal (e.g., interrupting discriminatory talk), communal (e.g., planning for potential deportations, joining organizations to advise teachers on how to respond to inequities), and political (e.g., extending citizenship rights). Methodologically, this article highlights the importance of research studies that move beyond the context of a single nation with study designs that are reflective of young people's lives across borders and varying sets of national systems and norms that often exclude them and their ways of knowing. As with young people's lived experiences and schooling, research on critical consciousness needs to cross geopolitical borders to truly understand how these specific inequities shape both experiences and the subaltern knowledge to contest them across multiple home countries.

Transborder youth's testimonios also provide important consejos, or advice, for educators and policymakers. Their words reveal how many young people were navigating fears related to familial deportation and undocumentedness, demonstrating the need for educators to learn about the realities of current immigration policies to better support students. A first step is including undocumentedness as a topic in teacher education programs. Just as educators learn to navigate and disrupt injustices related to racism, ableism, and other socially-constructed forms of discrimination, proactively learning about undocumentedness will better prepare educators to address and support students from undocumented families. As immigration policies are complex and shifting, educators can also rely on publicly available resources, such as those in colorincolorado.org or the Initiative on Immigration and Education at CUNY (https://www.cuny-iie.org/, accessed on 17 February 2023), to stay current on their educational implications. In their classrooms, educators can seek to overtly signal their support related to undocumentedness and com-

mitments that no human being is illegal, search for creative openings in the curriculum (see https://www.learningforjustice.org/, accessed on 17 February 2023), and ensure that they interrupt discriminatory behavior related to undocumentedness (see Gallo 2017). Even so, young people's fears related to deportation will not end until our immigration policies change, where pathways to citizenship or at least safeguards against deportation are extended across family members.

Finally, these findings reveal an important area for further research. Youth shared the constancy of fear they had felt since a young age in relation to their or their family members' undocumented status. Moreover, many of their experiences with family separations and other forms of inequities around undocumented status occurred in elementary school. This suggests that their critical consciousness formation began when they experienced these inequities as young children. This points to a need for further research to develop an understanding of what critical consciousness looks like in young transborder children navigating undocumented status in their families, which may be embodied and described differently from the measures often used with older students.

## 5. Conclusions

Ultimately, the young people we learn from here are using their critical consciousness to point us toward possible future worlds where they can access extended rights. Samuel and Alondra, who shared the ways that youth in Mexico were navigating the very real question of choosing between family or money, pushed us to imagine additional ways young people can build lives across both nations. When Samuel was asked if his critiques of U.S. and Mexican immigration systems meant that he was uninterested in living in the U.S., he was quick to laugh and counter this suggestion: "I AM interested", he said, elaborating the ways he imagined being able to relocate to the United States with access to U.S. papers. Indeed this would allow him to move freely between the U.S. and Mexico, eliminating the need to choose between economic prospects and seeing his family. Similarly, Abi's critiques of the present realities of our immigration systems in no way represented a desire to opt out—instead, she continued to seek and imagine a world where she would have the right to study her desired major at a university in the U.S., visit her grandparents in Mexico, and live free from the fear she has carried since she was a young girl. Abi's fear and Samuel's memories of family separation represent systemic inequities that are not inevitabilities. Similar to the transborder youth whose perspectives we highlight here, they continually seek ways to act towards building lives beyond the constraints imposed by national borders and hegemonizing epistemologies.

**Author Contributions:** Conceptualization, S.G. 50% and M.A.C. 50%; methodology, S.G. 100%; coding S.G. 50% M.A.C. 50%; formal analysis, S.G. 50% M.A.C. 50%; investigation, S.G. 100% writing—original draft preparation, S.G. 75%, M.A.C. 25%; writing—review and editing, S.G. 50% M.A.C. 50%; funding acquisition, S.G. 100%. All authors have read and agreed to the published version of the manuscript.

**Funding:** This research was funded by the Spencer Foundation and National Academy of Education Post-Doctoral Fellowship and the Fulbright García Robles Scholar Program in Mexico.

**Institutional Review Board Statement:** The studies were conducted in accordance with the Declaration of Helsinki, and approved by the Institutional Review of Ohio State University (protocol code 2016B0189 approved on 6/30/2016) and Rutgers University (protocol code 2020002891 on 12/02/2020) for studies involving humans.

**Informed Consent Statement:** Informed consent was obtained from all subjects involved in these studies.

**Data Availability Statement:** Data is not publicaly available due to privacy and ethical restrictions for human subjects research.

**Conflicts of Interest:** The authors declare no conflict of interest.

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
