# Peer review of "Documentation Status and Youth’s Critical Consciousness across Borders"

_socsci, doi:10.3390/socsci12040247_

Round 1

Reviewer 1 Report

This article is an exciting contribution to the scholarship on transborder youth, is very well organized, cited and has a powerful and clear methodology. The topic is urgent and important and the results shed light on a topic that is very timely. I recommend the editors to publish the article with no reservations.

Author Response

Dear Reviewer,

Thank you for your constructive feedback regarding our manuscript, ‘Documentation status and youth’s critical consciousness across borders.’ We appreciate your support of this important and timely research topic. In our revised draft we have added clarity to what this study contributes to critical consciousness research broadly and more precise implications for educators working with young people from mixed-status and undocumented families.

Best,

The Authors

Reviewer 2 Report

This article makes an important contribution. It is well-written, and organized. Moreover, the findings are richly detailed and succeeds in presenting the worldviews and experiences of a hidden population. I think it is ready for publication. 

Author Response

(The authors gave the same response as above.)

Reviewer 3 Report

Dear Authors,

  I am reviewing your article "Documentation Status and Youth’s Critical Consciousness Across Borders." I found the article to be interesting and well written. However, I feel that it could be improved by taking in cosiderations the following changes:   Abstract: This abstract provides an interesting overview of the article, but it could be improved by providing more detail and clarity on the specific research questions and methods used. Additionally, the article could benefit from more explicit language about how the research findings contribute to existing scholarship and how the article contributes to the broader field.   Introduction: This section of the article is well written and provides an interesting overview of the topic. The author does a good job of introducing the topic and setting up the research question. The author could improve the article by providing more information about the two qualitative studies that are being used to explore the topic. Additionally, the author could provide more detail about the specific experiences of transborder youth that relate to critical consciousness formation.   Literature review and teoreticall frame: This part of the article provides an excellent overview of the current literature and research related to transborder youth, schooling, and critical consciousness. The author provides an in-depth review of existing research, and does a great job of connecting this research to the current study. The author could improve this section by providing a more detailed explanation of how the current study builds upon existing research. Additionally, the author could include more examples from existing research to further illustrate the points they are making. The decision to mix framwork, literature review and methodology in one group is confusing.   Methods: This section of the article provides a thorough explanation of the methods used in the study. The author has done a good job of providing detailed information on the context of the study, the data collection and analysis, and the subjectivities of the researchers. To improve this section, the author could provide more information on the methods used to code the data and how the codes were used to draw meaningful themes from the data. Additionally, the author could provide more information on the ethical considerations taken into account when conducting the study.   Discussion: This part of the article provides a good overview of the research conducted and the findings. The authors could improve it by providing more information on the potential implications of their findings. For example, what can be done to address the fear that transborder youth experience in relation to their or their family members' undocumented status? What can be done to support transborder children in navigating undocumented status in their families? How can educators be better equipped to respond to the inequities experienced by transborder youth? what are the implications for policy making? These questions could be explored further in order to provide more insight into the implications of the research.Aditionally, the article could be improved by providing more detail about the results further discussing it with existing literature, that at the moment is lacking.   Best regards, Reviewer

Author Response

Dear Reviewer,

Thank you for your constructive feedback regarding our manuscript, ‘Documentation status and youth’s critical consciousness across borders.’ We appreciate your support of this important and timely research topic.

We have addressed your suggestions in the revised manuscript in the following ways:

Abstract: We have added information about the methodology and more explicitly named the implications to the larger field. Due to space constraints we did not add the research question to the abstract.

Introduction: We have added several sentences detailing each qualitative study in the introduction and summarizing the types of inequities related to undocumentedness that transborder youth in each study navigated.

Literature Review/Theoretical Frame: Throughout this section we have added more specificity on what this article contributes to the existing research. We also added more details regarding critical consciousness research within schools. Unfortunately, due to space constraints, we could not add further examples to other areas of the literature review.

Combination of framing/literature/methods: We agree that the co-existence of these 3 areas in the same section is a different format from most research articles that we write, and we have done this to follow the journal’s template. We hope that the sub-headings help distinguish each of these areas.

Methods: As suggested, we have added more details about our approach to coding and to our ethical considerations working with young people around undocumentedness. We also added a citation from Author 1’s work that addresses this topic more fully.

Discussion: We have majorly revised this section to explicitly answer the types of questions you named for educators and policy makers and added several resources that can be used to approach these challenging topics.

Contributions: Throughout the manuscript we have more explicitly named how the results from this research contribute to broader scholarship on critical consciousness.

Thank you again for your thoughtful feedback. We believe it has helped improve this manuscript.

Best,

The Authors